# Mediterranean Diet and Fatty Liver Risk in a Population of Overweight Older Italians: A Propensity Score-Matched Case-Cohort Study

**DOI:** 10.3390/nu14020258

**Published:** 2022-01-07

**Authors:** Luisa Lampignano, Rossella Donghia, Annamaria Sila, Ilaria Bortone, Rossella Tatoli, Sara De Nucci, Fabio Castellana, Roberta Zupo, Sarah Tirelli, Viviana Giannoccaro, Vito Guerra, Francesco Panza, Madia Lozupone, Mauro Mastronardi, Giovanni De Pergola, Gianluigi Giannelli, Rodolfo Sardone

**Affiliations:** 1Unit of Data Sciences and Technology Innovation for Population Health, National Institute of Gastroenterology “Saverio de Bellis”, Research Hospital, 70013 Bari, Italy; luisa.lampignano@irccsdebellis.it (L.L.); rossydonghia@gmail.com (R.D.); annamaria.sila@irccsdebellis.it (A.S.); ilariabortone@gmail.com (I.B.); rossella.tatoli@irccsdebellis.it (R.T.); sara.denucci@irccsdebellis.it (S.D.N.); fabio.castellana@irccsdebellis.it (F.C.); zuporoberta@gmail.com (R.Z.); sarah.tirelli@irccsdebellis.it (S.T.); vito.guerra@irccsdebellis.it (V.G.); 2Public Health Application Centre, 70043 Bari, Italy; vivianagiannoccaro@icloud.com; 3Neurodegenerative Disease Unit, Department of Basic Medicine, Neuroscience, and Sense Organs, University of Bari Aldo Moro, 11, 70125 Bari, Italy; f_panza@hotmail.com (F.P.); madia.lozupone@gmail.com (M.L.); 4Inflammatory Bowel Disease Unit, National Institute of Gastroenterology, “Saverio De Bellis” Research Hospital, 70013 Bari, Italy; mauro.mastronardi@irccsdebellis.it; 5Unit of Geriatrics and Internal Medicine, National Institute of Gastroenterology “Saverio de Bellis”, Research Hospital, 70013 Bari, Italy; gdepergola@libero.it; 6Scientific Direction, National Institute of Gastroenterology, “Saverio De Bellis” Research Hospital, 70013 Bari, Italy; gianluigi.giannelli@irccsdebellis.it

**Keywords:** NAFLD, Mediterranean diet, older, alcohol, meat, liver, fatty liver index (FLI)

## Abstract

Hepatic steatosis, often known as fatty liver, is the most common hepatic disease in Western countries. The latest guidelines for the treatment of nonalcoholic fatty liver disease emphasize lifestyle measures, such as changing unhealthy eating patterns. Using a propensity score-matching approach, this study investigated the effect of adhering to a Mediterranean diet (MedDiet) on fatty liver risk in an older population (≥65 years) from Southern Italy. We recruited 1.403 subjects (53.6% men, ≥65 years) who completed a food frequency questionnaire (FFQ) and underwent clinical assessment between 2015 and 2018. For the assessment of the liver fat content, we applied the Fatty Liver Index (FLI). To evaluate the treatment effect of the MedDiet, propensity score matching was performed on patients with and without FLI > 60. After propensity score-matching with the MedDiet pattern as treatment, we found a higher consumption of red meat (*p* = 0.04) and wine (*p* = 0.04) in subjects with FLI > 60. Based on the FLI, the inverse association shown between adherence to the MedDiet and the risk of hepatic steatosis shows that the MedDiet can help to prevent hepatic steatosis. Consuming less red and processed meat, as well as alcoholic beverages, may be part of these healthy lifestyle recommendations.

## 1. Introduction

Fatty liver (FL), also termed as hepatic steatosis (HS), is the most prominent liver disease in western countries [1], affecting up to 34 per cent of the general population and up to 70% per cent of obese people in industrialized countries [2,3,4]. This condition is characterized by pathological changes of the liver, featuring fatty infiltration of liver parenchyma (steatosis) that might induce an inflammatory status (nonalcoholic steatohepatitis, NASH) without excessive alcohol consumption [5].

Hepatic steatosis, with a fat content of more than 5 per cent of the liver volume, is the first identifiable level of non-alcoholic fatty liver disease (NAFLD) [1]. NAFLD can progress towards end-stage liver disease, including fibrosis, cirrhosis, and hepatocellular carcinoma [6]. Since hepatic steatosis raises the risk of metabolic syndrome, type 2 diabetes, and cardiovascular disease (CVD), prevention of this disorder is a public health concern [7]. Thus, European and American societies provide guidelines on managing nonalcoholic fatty liver disease [8,9], focusing on lifestyle interventions that mostly include weight reduction, physical exercise and the modification of unhealthy dietary habits. Interestingly, the dietary approach reflects prevailing patterns in the Mediterranean Sea area [8].

The Mediterranean Diet (MedDiet) refers to a set of eating patterns long practiced by people living along the Mediterranean Sea shores. It corresponds to a diet and lifestyle profile that was typical in the Mediterranean basin in the last century, characterized by a high intake of fruit, vegetables, and legumes, as well as a moderate intake of fish, the consumption of olive oil as the primary source of fats, and a low-to-moderate intake of red wine with meals [9].

Many studies have reported an inverse association between following a Mediterranean diet and the risk of CVD, stroke, cognitive impairment, and all-cause mortality [10]. This, and much other information from observational epidemiology, intervention trials and human experiments, form the rational basis for this diet [11].

Obesity and its comorbidities—primarily type 2 diabetes and hypertriglyceridemia—are likely accountable for the current FL epidemic in Italy, while alcohol consumption may provide a minor contribution [12,13]. Notably, this finding was also confirmed in other non-Italian studies [14,15].

Ageing is associated with a gradual increase in fat mass (FM), which usually reaches its peak at about 65 years in men, and afterwards in women. An augmentation in total FM can be independent of changes in body weight owing to a simultaneous decrease in muscle mass during ageing, known as sarcopenia [16]. This finding was endorsed by a longitudinal study conducted in Italy in community-dwelling older subjects [17]. A significant increase in the overall FM was observed despite maintaining the same weight [17]. A significant increase in waist circumference, with no weight gain, was observed in the same study after five years of follow-up [17]. Moreover, ageing is associated with a redistribution of adipose tissue (AT), independent of changes in body weight, even in stable-weight individuals [12]. In particular, visceral fat tends to increase, whereas subcutaneous fat decreases in other parts of the body (the abdomen, and especially the thighs and calves). Additionally, older people usually have a higher fat accumulation in the liver compared with younger people. This increased tissue store represents a discrepancy in the normal relationship between tissue uptake and liver fatty acid disposal due to a reduced sensitivity to insulin action [13]. It is noteworthy that nutrient may determine favorable or unfavorable changes in the amount, distribution, and function of body fat, even though evidence on this topic is limited and still quite uncertain, only a few studies having been conducted in cohorts with older subjects [18,19]. As nutrient intakes are closely interconnected and have synergistic effects, studies on the impact of single nutrients may not be able to reflect the complex influence of diet on AT. Anderson et al. declared that higher levels of nutritional status, quality of life and longevity in older adults might be correlated with a dietary pattern consisting of relatively high quantities of vegetables, fruit, whole grains, meat, fish, and low-fat dairy [20]. Further studies are required to determine the relationship between dietary patterns and the prevention and treatment of NAFLD, especially in older individuals, as this point is extremely relevant to the current public health scenario.

This study aimed to investigate the effect of adherence to a MedDiet on fatty liver in a cohort of older people from Southern Italy, using a propensity score-matching approach.

## 2. Materials and Methods

### 2.1. Study Population

Participants of the study were recruited among the residents of Castellana Grotte, Bari, Southern Italy. The sampling framework was based on the health registry office list on the 31st of December 2014, which contained 19675 individuals, of which 4021 were aged 65 years or more. The latter subjects were eligible for the “Salus in Apulia Study”, a public health program supported by the Italian Ministry of Health and Apulia Regional Government and carried out by the research hospital “Saverio De Bellis” belonging to the IRCCS research association. The potential study subjects overlapped with previous prospective MICOL studies started in 1981. All eligible subjects were invited to take part in the study, from 2014 until 2018, beginning with the MICOL participants. All participants signed an informed consent before their examination, and 1403 of them compiled a food frequency questionnaire (FFQ) and underwent clinical assessment between 2015 and 2018. The study was organized in accordance with the Helsinki Declaration of 1975, and adhered to the “Standards for Reporting Diagnostic Accuracy Studies” (STARD) guidelines. Available online: http://www.stard-statement.org/ (accessed on 30 November 2021). The reporting followed the “Strengthening the Reporting of Observational Studies in Epidemiology—Nutritional Epidemiology (STROBE-nut)” guidelines. Available online: https://www.strobe-nut.org/ (accessed on 30 November 2021).

### 2.2. Socio-Demographic, Lifestyle, Clinical, and Medical Characteristics

Smoking status was determined with the question, “Are you a current smoker?” Years of schooling were used to measure education levels. A YTON sphygmomanometer and a FARMAC-ZARBAN stethoscope were used to monitor blood pressure by professional nurses. Blood pressure was measured in a sitting position after rest. The final blood pressure values were the mean of the last two of three measurements. Body mass index (BMI) was determined as kg/m^2^. A Seca 220 stadiometer and a Seca 711 scale were used to measure height and weight. The narrowest section of the abdomen, or the area between the tenth rib and the iliac crest, was used to measure waist circumference (WC). Blood samples were obtained from the participants in the morning after overnight fast, and clinical biochemistry tests were performed. The glucose oxidase method (Sclavus, Siena, Italy) was used to quantify plasma glucose, whereas an automated colorimetric approach was used to determine plasma lipid concentrations (triglycerides, total cholesterol, and HDL cholesterol) (Hitachi; Boehringer Mannheim, Mannheim, Germany). LDL cholesterol was estimated through the Friedewald equation. Blood cell count, glutamyl oxaloacetic transaminase (GOT), glutamyl pyruvic transaminase (GPT) and gamma-glutamyl transferase (GGT) were measured using automatic enzyme procedures. An XT-2000i haematology analyser was used to determine the platelet count (Sysmex, Dasit, Cornaredo, Italy). Insulin resistance was determined with homeostasis model assessment–insulin resistance (HOMA-IR) [21].

In addition, the following pathological conditions were assessed. According to the American College of Cardiology American Heart Association criteria, hypertension status was determined as values >= 130/80 mmHg. Diabetes mellitus was categorized as fasting blood glucose (FBG) ≥ 126 mg/dL. Metabolic syndrome was diagnosed according to the International Diabetes Federation criteria [22]. The presence of stroke was confirmed through a medical history questionnaire conducted by a neurologist. Vascular dementia, dementia and depression were diagnosed according to the American Psychiatry Association criteria (DSM-IV-TR) [23]. Global cognitive function was assessed with the Mini-Mental State Examination (MMSE), and cognitive impairment was diagnosed at scores < 19 [24]. Sarcopenia was evaluated using the EWGSOP2 algorithm [25]. The Fried operational definition was used to determine the level of physical frailty. Peripheral age-related hearing loss was established, using pure tone audiometry on the frequencies of 0.5, 1, 2, and 4 kHz, as an average threshold greater than 40 dB hearing level in the better ear according to the World Health Organization (WHO) criteria of disabling hearing loss [26]. Vision loss was considered when visual acuity was less than 6/12 [27]. COPD was diagnosed by spirometry as a post-bronchodilator FEV1/FVC ratio < 0.70 [28]. Asthma was diagnosed according to Global Initiative for Asthma (GINA) guidelines on the medical history questionnaire and a spirometric assessment [29]. Lastly, multimorbidity status was defined as the copresence of two or more pathologies among the previously mentioned ones [30], as described elsewhere [31]. 

### 2.3. Dietary Assessments

The self-administered FFQ was divided into eleven sections, including foods with similar characteristics. At a later stage, the FFQ was validated against dietary records, and the results were analyzed to tailor the FFQ to our population [32]. The final questionnaire included 85 food items that best represented the regional diet, with some questions about edible fats. The 85 food items of the FFQ were regrouped and reduced to 29 food groups. One food group (cooking edible fats) could not be quantified and was therefore excluded from the study [33].

Moreover, we selected three widely mentioned diet indices, namely the Meddietscore, the DASH (dietary approaches to stop hypertension) diet index, and the MIND (Mediterranean-DASH intervention for neurodegenerative delay) diet index [34]. The congruency with the Mediterranean diet was evaluated according to the Meddietscore algorithm derived from the Greek Mediterranean diet pyramid [35]. The DASH diet index was developed as a result of a successful intervention trial that successfully lowered hypertension [36,37]. The DASH diet encouraged people to consume more fruits, vegetables, low-fat dairy, whole grains, poultry, fish, and nuts while minimizing red and processed meat, as well as added sweets. Total fat, saturated fat, and salt were all restricted in the original DASH diet [36]. Our study scoring of the DASH diet index was based on seven food groups and total fat, saturated fat, and sodium content. The DASH diet index could range from 0 (lowest) to 10 (highest concordance). The MIND diet index formulated by Morris et al. [34] is associated with a reduction in cognitive impairment. It comprises 15 dietary components: ten brain-healthy food groups (green leafy vegetables, other vegetables, nuts, berries, beans, whole grains, fish, poultry, olive oil and wine) and five brain-unhealthy food groups (red meats, butter and margarine, cheese, confectionery, and fast food). The MIND diet score ranged between 0 and 15. The FFQ and the three indices components have already been extensively described elsewhere [33].

### 2.4. Assessment of the Fatty Liver Status

To assess the fat content of the liver, we applied the algorithm of Bedogni et al. [38]. This group developed the “fatty liver index” (FLI), which predicts individual fatty liver (FL) status in the general population. The algorithm considers BMI, waist circumference, triglycerides and γ-GT derived from routine measurements in clinical practice [38].

### 2.5. Statistical Analysis

Subjects and food intake characteristics are reported as mean ± standard deviation (M ± SD) for continuous variables and as frequencies and percentages (%) for categorical variables. Normal distributions of quantitative variables were tested using the Kolmogorov–Smirnov test. For testing the associations between groups, the Chi-square test for categorical variables was used; when the variables were not normally distributed, the Wilcoxon rank-sum test was used for continuous variables.

To limit selection bias and to evaluate the treatment effect of the MedDiet, propensity score-matching (PSM) was performed to match patients with and without FLI, using the nearest neighbor-matching method with a 0.1 calliper width, without replacement, based on the following variables: age; gender; years of education; smoking; and alcohol consumption. Based on this score, we evaluated the effect of different dietary patterns on FLI as the outcome, calculating the average treatment effect (ATE).

All statistical analyses were performed using Stata statistical software, version STATA 16, 2019 (StataCorp LP, College Station, TX, USA).

## 3. Results

Characteristics of participants, according to FLI score categories, are shown in Table 1. Among 1403 (53.6% men) participants who were available and analyzed before propensity score-matching, 52.2% (*n* = 732) were classified as having a higher risk of NAFLD (FLI > 60). Participants with FLI >60 tended to be older (*p* < 0.0001), current smokers (*p* < 0.0001), more sedentary (*p* < 0.0001), had higher BMI (*p* < 0.0001) and higher WC (*p* < 0.0001) than participants with FLI < 60. Serum levels of glucose, total cholesterol, triglycerides, GPT, GGT (*p* < 0.0001), GOT (*p* = 0.001) and platelets count (*p* = 0.0002) were higher in subjects with a higher risk of NAFLD, whereas HDL serum levels were lower (*p* < 0.0001). HOMA-IR was higher in people with FLI > 60. According to the diagnosis, we found a higher prevalence of hypertension (*p* = 0.003), diabetes (*p* = 0.05), metabolic syndrome (*p* < 0.001), dementia (*p* = 0.01) and multimorbidity status (*p* = 0.003) in the group with a higher risk of NAFLD. 

Table 2 shows the average amounts of food groups and macronutrients and micronutrients consumed by the study subjects divided by the FLI score. 

In the unmatched cohort, we found a lower consumption of low-fat dairy (*p* = 0.04), nuts (*p* = 0.002), and coffee (*p* = 0.05) in the FLI > 60 group.

Table 3 shows the ATE of the three different dietary indices (MedDiet Score, DASH index and MIND index) on FLI. We found that a MedDiet Score ≥ of 30.00 (cut-off derived from the median) was inversely associated with FLI (β = −0.07, 95% CI: −0.13 to −0.01). Hence, people who showed an adherence above the median value to the MedDiet had, on average, a 7% lower FLI.

To assess the effect of foods nutrients and micronutrients, independently of age, gender, years of education, smoking and alcohol consumption, considering this group as treated with MedDiet (score above 27.5), the characteristics of participants according to FLI score after propensity score-matching with the MedDiet pattern as treatment, are shown in Table 4. This step generated 343 cases and 284 controls, showing significant differences in smoking habits (*p* < 0.001), years of education (*p* = 0.04), glucose (*p* = 0.04), triglycerides (*p* = 0.002) and GGT (*p* = 0.02) serum levels, and multimorbidity status (*p* = 0.04).

Table 5 shows the average amounts of food groups and macronutrients and micronutrients consumed by the subjects divided by FLI score, after propensity score-matching with the MedDiet pattern as treatment. We found a higher consumption of red meat (*p* = 0.04) and wine (*p* = 0.04) and a higher intake of alcohol (*p* = 0.05) in subjects with FLI > 60.

## 4. Discussion

In our study, using propensity score-matching to estimate the effect of three nutritional treatments (MedDiet, DASH and MIND) on fatty liver in an older population from Southern Italy, we found an inverse association between a higher adherence to MedDiet and the risk of fatty liver. 

The hypothesis that the conventional MedDiet, a plant-based diet high in unsaturated fat, is helpful in the prevention and treatment of metabolic syndrome, CVD, and their risk factors, is endorsed on a solid scientific basis [9,11,39,40,41,42]. Therefore, recent findings, mainly from epidemiological research, indicate that adherence to the MedDiet is also beneficial against NAFLD [9]. A clinical trial in diabetic patients with NAFLD showed a decrease in liver steatosis after a 6-week dietary intervention using a MedDiet compared with a control diet [43]. In 2008, a trial investigated the potential effect of three different diets on the liver enzymes of subjects with obesity and type 2 diabetes [44]. Patients were randomized to a modified MedDiet, a diet suggested by the American Diabetes Association, and a low glycemic index diet. The three diets had a similar content of total fat, but different ratios of carbohydrates and MUFA; the MedDiet had a higher proportion of unsaturated fat and a lower percentage of energy coming from carbohydrates than the other two diets, to ensure all low-glycemic index meals. At 6 and 12 months of follow-up, ALT levels had decreased more significantly in the modified MedDiet arm than in the other two dietary profiles, (mean values reduced by about 5 U/L), even after adjustment for some traditional risk factors, including change in body mass, triglycerides, and insulin resistance from the baseline. Despite the small sample size and the clinical evaluation limited to hepatic enzymes, this trial reported for the first time that a MedDiet might lower ALT levels, and that this effect was independent from weight loss or decreases in other blood biomarkers [44].

Later, a Greek study involving a cohort of more than 3000 subjects confirmed the beneficial effect of a MedDiet on hepatic enzymes [45]. The authors analyzed adherence to the Mediterranean diet using the same score of our study.

Furthermore, a randomized, crossover dietary intervention trial by Ryan et al. provided more evidence of the therapeutic role of a MedDiet on fatty liver [43]. All patients had biopsy-proven NAFLD and were randomized to either a MedDiet or a control diet for a duration of 6 weeks, interspersed by a wash-out period. At the conclusion of the intervention period, a significant decrease in liver fat content was revealed with MRI after the MedDiet protocol compared with the control group, although weight loss was comparable between the two groups. Moreover, patients showed improved insulin sensitivity and circulating insulin levels only after the MedDiet phase. Interestingly, no significant variations in AST and ALT levels were observed.

In 2014, adherence to a MedDiet (estimated through the MedDiet Score) was examined in association with NAFLD severity [46], assessed through transient elastography and liver biopsies. A strong negative correlation between the MedDiet Score and ALT, insulin levels, fibrosis, and steatosis severity was evidenced in patients with NAFLD. 

As a model for healthier eating, this dietary pattern has been advocated worldwide. It has been documented to lead to a favorable health status and provide an optimum intake of salutary nutrients to prevent chronic degenerative diseases [47]. The benefits of such dietary habits can be expressed in terms of NAFLD prevention through many mechanisms that can differ, ranging from an appropriate dietary strategy for weight loss, to a model diet that is abundant in some beneficial nutrients such as MUFA and vitamins, with the inclusion of olive oil as the key source of fat [48]. Each of these considerations likely contributes, overall, to establishing the protective and therapeutic role of the MedDiet in NAFLD.

It is clear that, in our study, the positive impact of the MedDiet is due to the synergy of several food groups that characterize this diet, rather than to individual foods. In particular, showing the same effect as the MedDiet, the difference between older subjects who had a high risk of having NAFLD and older individuals who did not, was the higher consumption of red meat and alcohol.

Meat, in general, provides nutrients that are beneficial to human health, such as protein, iron, zinc, and vitamin B12 [49]. On the other hand, it provides saturated fatty acids (SFA) and cholesterol, all of which are detrimental to subjects with NAFLD [50,51,52], together with other plausibly damaging compounds such as heme iron [53] and sodium [54]. Indeed, frequent meat consumption has been proven to be associated with oxidative stress [55], metabolic syndrome [56], IR and type 2 diabetes [56,57,58]. Red meat, in particular, has been linked to an increased risk of mortality due to chronic liver disease and hepatocellular carcinoma [59]. An association between consuming meat and NAFLD has been identified in some studies [60,61,62].

A variety of epidemiologic studies worldwide have shown that a mild alcohol intake can help prevent the development of NAFLD, mainly by enhancing peripheral insulin tolerance [63].

When serum transaminases production is used as the endpoint, epidemiological data also confirm the beneficial effect of a mild alcohol intake on the liver. In a study involving more than 1000 participants without hepatic diseases, the authors showed that light (70–140 g per week) to moderate (140–280 g per week) alcohol consumption was associated with reduced serum levels of transaminases, compared with the controls [64]. However, when testing for covariates such as BMI, drinking alcohol in the higher range (more than three drinks a day) was associated with increases in ALT and AST, as predicted [65]. Despite weaknesses in some of these epidemiological study research designs, current evidence suggests that mild to moderate drinking can protect healthy people from diabetes and hepatic steatosis [63].

However, in the elderly, sensitivity to the effects of alcohol increases due to physiological and metabolic changes in the body over the years [66]. From the age of about 50 years, the quantity of water present in the body decreases, and alcohol is diluted in a smaller amount of liquid. This means that, when ingesting the same amount of alcohol, the alcohol content in the blood is higher, and the effects are more pronounced. This phenomenon adds to the reduced functioning of some organs, such as the liver and kidneys, which can no longer fully perform the function of inactivating the toxic action of alcohol, and allowing its elimination from the body. It should also be considered that older people often suffer from balance problems, due to weakening muscles and reduced mobility. Alcohol consumption can, therefore, aggravate the situation, facilitating falls and fractures. It must also be added that alcohol interferes with the use of drugs, taken daily by the elderly, in most cases. Therefore, in old age, even a moderate consumption of alcohol can cause health problems [67]. The Italian Guidelines for a healthy diet, a review from 2018 presented by the Food and Nutrition Research Center–CREA), advise the elderly not to exceed the limit of 12 g of alcohol per day, equal to 1 Alcohol Unit (330 mL of beer, 125 mL of wine or 40 mL of spirits) without distinction between men and women [68].

Lastly, in our study population, smoking status and education years were different in PSM-matched cohorts. Since the matching was based on partial sampling—not 1:1 but on the nearest neighbors—it was not sensitive to large differences in frequency, as in the case of smoking and education. The effect of smoking habits and education was very important in the diet, because usually smokers have an unhealthy lifestyle [69] and a lower level of education [70]. This concept is also valid in older subjects, even though the education level was low in both groups of our study (Table 4). Therefore, they should always be considered as confounding factors in association models. The strengths of this study include its well-defined population, the standardized and clinically based assessments to measure liver fat deposition, and the use of a validated FFQ to obtain dietary information. 

In this study, several limitations were present: (1) FFQ is a dietary assessment tool affected by a strong recall bias, especially for older subjects. Despite this, FFQs remain the most widely used dietary evaluation tool in cohort studies. (2) The cross-sectional nature did not reveal a clear directionality of the association, resulting in a high risk of reverse-causality bias. (3) We did not use a gold standard method such as ultrasonography to evaluate hepatic and visceral fat values.

## 5. Conclusions

The inverse association pointed out between adherence to the MedDiet and the risk of hepatic steatosis based on the FLI suggests that the MedDiet can play a role in preventing NAFLD. In particular, consuming less red and processed meat and alcoholic beverages may be part of healthy lifestyle recommendations readily proposed by primary care doctors in their general practice. To better understand the causal effect of a particular food consumption on liver fat storage, longitudinal population-based studies in larger sample sizes are needed.

## Figures and Tables

**Table 1 nutrients-14-00258-t001:** Baseline and clinical characteristics of patients in the unmatched cohort.

Variables *	Unmatched Cohort		
	FLI		
	≤60 (*n* = 671)	>60 (*n* = 732)	*p* ^^^
Gender (%)			0.97 ^§^
Male	360 (53.65)	392 (53.55)	
Female	311 (46.35)	340 (46.45)	
Age (years)	77.76 ± 7.61	79.47 ± 8.52	0.001
Smoking (Yes) (%)	110 (16.39)	273 (37.30)	<0.001 ^§^
Education (years)	7.55 ± 5.23	8.26 ± 6.02	0.10
Physical Activity (<2) (%)	253 (40.03)	372 (54.31)	<0.001 ^§^
Systolic Pressure (mmHg)	131.97 ± 14.38	134.50 ± 14.34	0.002
Diastolic Pressure (mmHg)	78.22 ± 8.18	78.53 ± 7.74	0.59
BMI (Kg/m^2^)	26.48 ± 2.90	31.16 ± 4.18	<0.0001
BMI (Kg/m^2^) (%)			<0.001 ^§^
<25	221 (32.94)	33 (4.51)	
≥25	450 (67.06)	699 (95.49)	
Waist (cm)	97.28 ± 8.01	108.39 ± 9.40	<0.0001
Waist (cm) by Gender			
Male	97.95 ± 7.27	109.54 ± 8.90	<0.0001
Female	96.50 ± 8.73	107.08 ± 9.79	<0.0001
Glucose (mg/dL)	105.80 ± 25.81	121.55 ± 41.67	<0.0001
HOMA-IR (mg/dL)	1.98 ± 1.69	3.24 ± 2.98	<0.0001
Total Cholesterol (mg/dL)	184.68 ± 38.67	196.18 ± 41.10	<0.0001
Triglycerides (mg/dL)	98.51 ± 52.08	180.03 ± 97.96	<0.0001
HDL (mg/dL)	51.91 ± 14.71	48.83 ± 14.83	<0.0001
LDL (mg/dL)	114.52 ± 35.37	114.36 ± 32.64	0.85
GOT (U/L)	28.56 ± 22.14	33.64 ± 30.28	0.001
GPT (U/L)	22.67 ± 16.23	28.16 ± 22.41	<0.0001
GGT (U/L)	19.30 ± 15.28	47.09 ± 44.11	<0.0001
Platelets count	223.72 ± 68.76	237.10 ± 70.78	0.0002
Hypertension (Yes) (%)	453 (67.51)	546 (74.59)	0.003 ^§^
Diabetes (Yes) (%)	77 (11.48)	110 (15.03)	0.05 ^§^
Metabolic Syndrome (Yes) (%)	50 (7.45)	118 (16.12)	<0.001 ^§^
Stroke (Yes) (%)	13 (1.94)	18 (2.46)	0.51 ^§^
Vascular Dementia (%)	2 (0.30)	2 (0.27)	0.93 ^§^
Dementia (Yes) (%)	34 (5.07)	62 (8.47)	0.01 ^§^
Depression (Yes) (%)	68 (10.13)	93 (12.70)	0.13 ^§^
MMSE < 19 (Yes) (%)	29 (4.32)	46 (6.28)	0.10 ^§^
Sarcopenia (Yes) (%)	71 (10.58)	62 (8.47)	0.18 ^§^
Physical Frailty (Yes) (%)	129 (19.23)	131 (17.90)	0.52 ^§^
ARHL (Yes) (%)	139 (20.72)	169 (23.09)	0.28 ^§^
Vision Loss (Yes) (%)	25 (3.73)	26 (3.55)	0.86 ^§^
COPD (Yes) (%)	118 (17.59)	140 (19.13)	0.46 ^§^
Asthma (Yes) (%)	61 (9.09)	70 (9.56)	0.76 ^§^
Multimorbidity (>1) (Yes) (%)	369 (54.99)	460 (62.84)	0.003 ^§^

* Reported as mean and standard deviation (Mean ± SD) for continuous values and percentage for categorical variables. Abbreviations: FLI, fatty liver index; BMI, body mass index; HOMA-IR, homeostatic model assessment for insulin resistance; HDL, high-density lipoproteins; LDL, low-density lipoproteins; GOT, glutamic oxaloacetic transaminase; GPT, glutamic pyruvic transaminase; GGT, gamma-glutamyl transferase; ARHL, age-related hearing loss; COPD, chronic obstructive pulmonary disease; MMSE, mini-mental state. Examination. ^^^ Wilcoxon rank-sum test (Mann–Whitney), ^§^ Chi-Square test, where necessary.

**Table 2 nutrients-14-00258-t002:** Food, micro and macro intake in the unmatched cohort.

Variables *	Unmatched Cohort		
	FLI		
	≤60 (*n* = 671)	>60 (*n* = 732)	*p* ^^^
Food Groups			
Dairy	104.41 ± 113.22	105.91 ± 107.37	0.38
Low Fat Dairy	105.25 ± 107.11	97.52 ± 109.10	0.04
Eggs	8.49 ± 9.45	7.94 ± 8.67	0.31
White Meat	24.68 ± 27.69	28.22 ± 44.06	0.07
Red Meat	22.38 ± 25.81	23.69 ± 26.59	0.14
Processed Meat	14.58 ± 15.10	16.20 ± 24.65	0.12
Fish	25.78 ± 25.17	26.78 ± 54.22	0.69
Seafood/Shellfish	9.09 ± 12.89	11.18 ± 34.91	0.28
Leafy Vegetables	58.66 ± 60.33	60.90 ± 70.39	0.96
Fruiting Vegetables	95.69 ± 77.16	95.20 ± 87.63	0.30
Root Vegetables	11.67 ± 26.58	12.52 ± 28.74	0.66
Other Vegetables	83.40 ± 80.45	81.34 ± 83.66	0.45
Legumes	39.12 ± 34.28	37.38 ± 27.48	0.23
Potatoes	13.55 ± 18.38	13.25 ± 19.56	0.23
Fruits	629.41 ± 550.79	605.42 ± 510.80	0.63
Nuts	8.01 ± 15.69	6.63 ± 15.82	0.002
Grains	157.82 ± 106.63	154.54 ± 108.05	0.43
Olives and Vegetable Oil	52.67 ± 36.07	51.33 ± 38.76	0.06
Sweets	22.64 ± 28.04	22.93 ± 39.36	0.14
Sugary	10.69 ± 15.49	10.46 ± 25.29	0.12
Juices	7.22 ± 21.25	6.13 ± 20.18	0.44
Caloric Drinks	7.14 ± 43.08	9.87 ± 59.99	0.67
Ready to Eat Dish	24.01 ± 37.21	32.76 ± 54.78	0.35
Coffee	48.37 ± 29.81	45.64 ± 29.78	0.05
Wine	118.76 ± 159.54	125.59 ± 167.17	0.98
Beer	17.00 ± 67.02	21.76 ± 77.45	0.29
Spirits	1.32 ± 4.55	1.67 ± 6.16	0.80
Water	657.40 ± 302.65	663.81 ± 298.00	0.60
Micro-Nutrients			
Na^+^	1530.14 ± 861.60	1547.66 ± 1131.46	0.97
K^+^	3438.53 ± 1757.71	3385.37 ± 1845.44	0.67
Fe	11.08 ± 5.03	11.18 ± 5.94	0.99
Ca^++^	878.10 ± 526.06	882.73 ± 564.11	0.63
P	1133.43 ± 500.70	1144.40 ± 629.48	0.53
B_1_	0.83 ± 0.37	0.84 ± 0.48	0.79
B_2_	1.43 ± 0.75	1.42 ± 0.76	0.91
PP	1.43 ± 0.75	1.42 ± 0.76	0.91
Vitamin A	1192.58 ± 1785.23	1215.04 ± 1711.93	0.69
Vitamin C	187.09 ± 130.17	181.44 ± 130.47	0.55
Macro-Nutrients			
H_2_O	1954.08 ± 744.49	1947.36 ± 761.35	0.93
Proteins	66.78 ± 28.97	67.84 ± 41.99	0.70
Lipids	84.72 ± 38.62	83.73 ± 45.73	0.12
Total Carbohydrates	471.32 ± 215.41	458.12 ± 228.93	0.13
Total Fibers	74.51 ± 40.69	71.91 ± 39.67	0.29
Saturated Fatty Acids	21.73 ± 10.73	21.99 ± 12.46	0.59
MUFAs	44.51 ± 24.87	43.68 ± 27.91	0.06
PUFAs	9.40 ± 4.37	9.19 ± 5.57	0.03
Cholesterol	189.92 ± 117.51	196.39 ± 154.89	0.76
Alcohol	13.13 ± 17.25	14.03 ± 18.53	0.91
Alcohol >20 (Yes) (%)	128 (19.08)	156 (21.31)	0.30 ^§^
Kcal	2037.62 ± 719.38	2013.98 ± 877.32	0.15
KJ	8529.16 ± 3011.50	8430.23 ± 3672.43	0.15

* Reported as mean and standard deviation (Mean ± SD) for continuous values and percentage for categorical variables. All food groups were calculated on quantity of daily consumption. Abbreviations: FLI, fatty liver index; MUFAs, monounsaturated fatty acids; PUFAs, polyunsaturated fatty acids; Kcal, kilocalories; KJ, kilojoules. ^^^ Wilcoxon rank-sum test (Mann–Whitney), ^§^ Chi-Square test, where necessary.

**Table 3 nutrients-14-00258-t003:** Average Treatment Effects (ATE) on FLI.

Dietary Pattern	β	se(β)	*p*-Value	C.I. (95%)
MedDiet	−0.07	0.03	0.02	−0.13 to −0.01
DASH	−0.06	0.03	0.07	−0.12 to 0.005
MIND	−0.05	0.03	0.15	−0.11 to 0.02

Abbreviations: β, Coefficient; se(β), Standard Error; C.I., 95% Confidence Intervals; MedDiet (Mediterranean diet); DASH (dietary approaches to stop hypertension); MIND (Mediterranean-DASH intervention for neurodegenerative delay).

**Table 4 nutrients-14-00258-t004:** Baseline and clinical characteristics of patients in the matched cohort.

Variables *	Matched Cohort		
	FLI		
	≤60 (*n* = 284)	>60 (*n* = 343)	*p* ^^^
Gender (%)			0.27 ^§^
Male	144 (50.70)	189 (55.10)	
Female	140 (49.30)	154 (44.90)	
Age (years)	77.90 ± 7.84	79.09 ± 8.30	0.10
Smoking (Yes) (%)	51 (17.96)	117 (34.11)	<0.001 ^§^
Education (years)	7.33 ± 5.23	8.40 ± 5.93	0.04
Physical Activity (<2) (%)	125 (46.13)	161 (50.16)	0.33 ^§^
Systolic Pressure (mmHg)	133.03 ± 14.04	133.53 ± 13.47	0.63
Diastolic Pressure (mmHg)	78.29 ± 7.48	78.50 ± 7.81	0.86
BMI (Kg/m^2^)	28.80 ± 4.59	29.91 ± 4.41	0.53
BMI (Kg/m^2^) (%)			0.67 ^§^
<25	56 (19.72)	63 (18.37)	
≥25	228 (80.28)	280 (81.63)	
Waist (cm)	102.42 ± 9.65	103.51 ± 11.21	0.20
Waist (cm) by Gender			
Male	102.71 ± 9.39	104.31 ± 10.47	0.14
Female	102.12 ± 9.94	102.52 ± 12.02	0.80
Glucose (mg/dL)	110.35 ± 33.29	114.68 ± 33.96	0.04
HOMA-IR (mg/dL)	2.57 ± 2.71	2.57 ± 2.52	0.34
Total Cholesterol (mg/dL)	186.13 ± 39.65	192.53 ± 40.57	0.08
Triglycerides (mg/dL)	125.12 ± 81.50	142.54 ± 85.48	0.002
HDL (mg/dL)	51.02 ± 15.02	49.81 ± 13.91	0.37
LDL (mg/dL)	111.68 ± 33.65	116.88 ± 34.96	0.08
GOT (U/L)	28.91 ± 22.63	32.72 ± 27.50	0.20
GPT (U/L)	24.69 ± 16.00	27.29 ± 24.28	0.60
GGT (U/L)	30.23 ± 34.90	34.14 ± 36.64	0.02
Platelets count	225.96 ± 64.06	229.28 ± 69.90	0.67
Hypertension (Yes) (%)	193 (67.96)	255 (74.34)	0.08 ^§^
Diabetes (Yes) (%)	35 (12.32)	43 (12.54)	0.94 ^§^
Metabolic Syndrome (Yes) (%)	33 (11.62)	48 (13.99)	0.38 ^§^
Stroke (Yes) (%)	6 (2.11)	6 (1.75)	0.74 ^§^
Vascular Dementia (%)	1 (0.35)	0 (0.00)	0.27 ^§^
Dementia (Yes) (%)	12 (4.23)	18 (5.25)	0.55 ^§^
Depression (Yes) (%)	29 (10.21)	49 (14.29)	0.12 ^§^
MMSE < 19 (Yes) (%)	9 (3.17)	14 (4.08)	0.54 ^§^
Sarcopenia (Yes) (%)	26 (9.15)	33 (9.62)	0.84 ^§^
Physical Frailty (Yes) (%)	54 (19.01)	67 (19.53)	0.87 ^§^
ARHL (Yes) (%)	58 (20.42)	78 (22.74)	0.48 ^§^
Vision Loss (Yes) (%)	6 (2.11)	10 (2.92)	0.53 ^§^
COPD (Yes) (%)	43 (15.14)	71 (20.70)	0.07 ^§^
Asthma (Yes) (%)	23 (8.10)	36 (10.50)	0.31 ^§^
Multimorbidity (>1) (Yes) (%)	153 (53.87)	213 (62.10)	0.04 ^§^

* Reported as mean and standard deviation (Mean ± SD) for continuous values and percentage for categorical variables. Abbreviations: FLI, fatty liver index; BMI, body mass index; HOMA-IR, homeostatic model assessment for insulin resistance; HDL, high-density lipoproteins; LDL, low-density lipoproteins; GOT, glutamic oxaloacetic transaminase; GPT, glutamic pyruvic transaminase; GGT, gamma-glutamyl transferase; ARHL, age-related hearing loss; COPD, chronic obstructive pulmonary disease; MMSE, mini-mental State. Examination. ^^^ Wilcoxon rank-sum test (Mann–Whitney), ^§^ Chi-Square test, where necessary.

**Table 5 nutrients-14-00258-t005:** Food, micro and macro intake in the matched cohort.

Variables *	Matched Cohort		
	FLI		
	≤60 (*n =* 284)	>60 (*n =* 343)	*p* ^^^
Food Groups			
Dairy	108.74 ± 121.47	113.78 ± 123.10	0.52
Low Fat Dairy	108.91 ± 109.63	115.78 ± 115.25	0.48
Eggs	9.80 ± 11.27	9.52 ± 9.04	0.53
White Meat	28.73 ± 23.97	35.53 ± 53.14	0.18
Red Meat	21.40 ± 14.97	27.98 ± 36.79	0.04
Processed Meat	14.72 ± 14.16	19.33 ± 34.32	0.06
Fish	31.50 ± 28.30	36.58 ± 77.22	0.97
Seafood/Shellfish	10.74 ± 11.61	14.02 ± 47.78	0.18
Leafy Vegetables	81.30 ± 79.71	85.69 ± 81.87	0.18
Fruiting Vegetables	131.20 ± 94.95	130.04 ± 93.53	0.65
Root Vegetables	15.42 ± 27.09	16.93 ± 30.71	0.14
Other Vegetables	117.21 ± 99.56	109.49 ± 96.89	0.40
Legumes	51.94 ± 47.53	48.15 ± 27.49	0.94
Potatoes	16.94 ± 18.45	20.56 ± 28.96	0.08
Fruits	802.60 ± 618.56	760.48 ± 526.03	0.74
Nuts	11.59 ± 20.27	9.49 ± 18.30	0.85
Grains	182.10 ± 110.15	175.16 ± 112.36	0.26
Olives and Vegetable Oil	55.79 ± 37.81	60.16 ± 40.16	0.09
Sweets	28.00 ± 36.63	25.23 ± 39.34	0.81
Sugary	13.92 ± 38.56	11.84 ± 19.78	0.73
Juices	6.35 ± 21.44	8.57 ± 26.16	0.33
Caloric Drinks	3.55 ± 10.20	12.42 ± 64.56	0.98
Ready to Eat Dish	35.70 ± 35.70	42.84 ± 78.51	0.60
Coffee	46.48 ± 29.96	48.89 ± 30.61	0.31
Wine	96.48 ± 157.82	130.93 ± 188.31	0.04
Beer	17.43 ± 66.63	20.72 ± 77.18	0.69
Spirits	1.37 ± 4.92	1.21 ± 4.29	0.97
Water	718.18 ± 281.97	692.55 ± 305.98	0.27
Micro-Nutrients			
Na^+^	1746.09 ± 833.28	1901.85 ± 1491.34	0.30
K^+^	4255.60 ± 1953.77	4258.50 ± 1984.92	0.80
Fe	13.07 ± 5.15	13.73 ± 7.04	0.34
Ca^++^	990.01 ± 573.63	1050.74 ± 591.58	0.18
P	1274.45 ± 507.97	1372.52 ± 764.93	0.12
B_1_	0.99 ± 0.37	1.04 ± 0.61	0.84
B_2_	1.66 ± 0.72	1.75 ± 0.90	0.29
PP	1.66 ± 0.72	1.75 ± 0.90	0.29
Vitamin A	1411.61 ± 868.50	1607.07 ± 2376.31	0.15
Vitamin C	240.52 ± 148.46	234.61 ± 138.90	0.88
*Macro-Nutrients*			
H_2_O	2266.55 ± 808.90	2272.44 ± 770.32	0.51
Proteins	76.50 ± 27.22	82.04 ± 53.96	0.40
Lipids	92.08 ± 38.78	97.82 ± 48.32	0.12
Total Carbohydrates	556.42 ± 220.89	546.21 ± 236.13	0.46
Total Fibers	94.03 ± 43.58	91.61 ± 40.44	0.67
Saturated Fatty Acids	23.52 ± 11.41	25.22 ± 13.25	0.12
MUFAs	47.26 ± 25.43	50.11 ± 27.67	0.09
PUFAs	10.85 ± 4.69	11.20 ± 6.39	0.53
Cholesterol	214.56 ± 122.31	234.38 ± 191.61	0.14
Alcohol	10.84 ± 17.42	14.48 ± 20.58	0.05
Alcohol >20 (Yes) (%)	42 (14.79)	80 (23.32)	0.007 ^§^
Kcal	2290.24 ± 701.19	2370.45 ± 973.63	0.60
KJ	9586.85 ± 2935.47	9922.48 ± 4075.39	0.60

* Reported as mean and standard Deviation (Mean ± SD) for continuous values and percentage for categorical variables. All food groups were calculated on quantity daily consumption. Abbreviations: FLI, fatty liver index; MUFAs, monounsaturated fatty acids; PUFAs, polyunsaturated fatty acids; Kcal, kilocalories; KJ, kilojoules. ^^^ Wilcoxon rank-sum test (Mann–Whitney), ^§^ Chi-Square test, where necessary.

## Data Availability

The datasets used and/or analyzed during the current study are available from the corresponding author on reasonable request.

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
