# Peer review of "Mediterranean Diet and Fatty Liver Risk in a Population of Overweight Older Italians: A Propensity Score-Matched Case-Cohort Study"

_nutrients, 2022, doi:10.3390/nu14020258_

Round 1

Reviewer 1 Report

Authors performed PSM matching using well-characterized Southern Italian elderly cohort to assess the Mediterranean diet for the risk of fatty liver. As the results, the Mediterranean diet might have preventive effect for hepatic steatosis. This study is interesting and is useful for clinicians. In present cohort, smoking status and education years were different in PSM-matched cohorts. Authors should discuss their significance in the results.   

Author Response

Firstly, we would like to thank the reviewer for their suggestions and the precious time spent reviewing our manuscript.

We implemented the discussion in lines 379-386 "Moreover, in our study population, smoking status and education years were different in PSM-matched cohorts. Since the matching is based on partial sampling not 1:1 but on nearest neighbours, it is not sensitive to large differences in frequency, as in the case of smoking and education. The effect of smoking habits and education is very important in the diet because usually smokers have an unhealthy lifestyle and a lower level of education, and this concept is also valid in older subjects even if the education level is low in both groups of our study (Table 4). Therefore, they should always be considered as confounding factors in association models."

Reviewer 2 Report

In my opinion, the manuscript was prepared well, so I suggest accepting it after minor text editing. 
•    The Introduction Section explains the design of the study. The Authors well justify the research topic. Minor language fixes required. 
•    The Descriptions of the results were correct.
•    The presented Tables were prepared precisely.
•    The Conclusions were well formulated.

Author Response

Firstly, we would like to thank the reviewer for the precious time spent reviewing our manuscript.
We have corrected the language errors in the introduction.